# Bandit Smooth Convex Optimization: Improving the Bias-Variance Tradeoff

**Ofer Dekel**
Microsoft Research
Redmond, WA
oferd@microsoft.com

**Ronen Eldan**
Weizmann Institute
Rehovot, Israel
roneneldan@gmail.com

**Tomer Koren**
Technion
Haifa, Israel
tomerk@technion.ac.il

## Abstract

Bandit convex optimization is one of the fundamental problems in the field of online learning. The best algorithm for the general bandit convex optimization problem guarantees a regret of $\widetilde{O}(T^{5/6})$, while the best known lower bound is $\Omega(T^{1/2})$. Many attempts have been made to bridge the huge gap between these bounds. A particularly interesting special case of this problem assumes that the loss functions are smooth. In this case, the best known algorithm guarantees a regret of $\widetilde{O}(T^{2/3})$. We present an efficient algorithm for the bandit smooth convex optimization problem that guarantees a regret of $\widetilde{O}(T^{5/8})$. Our result rules out an $\Omega(T^{2/3})$ lower bound and takes a significant step towards the resolution of this open problem.

## 1 Introduction

Bandit convex optimization [11, 5] is the following online learning problem. First, an adversary privately chooses a sequence of bounded and convex loss functions $f_1, \ldots, f_T$ defined over a convex domain $K$ in $d$-dimensional Euclidean space. Then, a randomized decision maker iteratively chooses a sequence of points $x_1, \ldots, x_T$, where each $x_t \in K$. On iteration $t$, after choosing the point $x_t$, the decision maker incurs a loss of $f_t(x_t)$ and receives *bandit feedback*: he observes the value of his loss but he does not receive any other information about the function $f_t$. The decision maker uses the feedback to make better choices on subsequent rounds. His goal is to minimize *regret*, which is the difference between his loss and the loss incurred by the best fixed point in $K$. If the regret grows sublinearly with $T$, it indicates that the decision maker's performance improves as the length of the sequence increases, and therefore we say that he is learning.

Finding an optimal algorithm for bandit convex optimization is an elusive open problem. The first algorithm for this problem was presented in Flaxman et al. [11] and guarantees a regret of $\mathcal{R}(T) = \widetilde{O}(T^{5/6})$ for any sequence of loss functions (here and throughout, the asymptotic $\widetilde{O}$ notation hides a polynomial dependence on the dimension $d$ as well as logarithmic factors). Despite the ongoing effort to improve on this rate, it remains the state of the art. On the other hand, Dani et al. [9] proves that for any algorithm there exists a worst-case sequence of loss functions for which $\mathcal{R}(T) = \Omega(T^{1/2})$, and the gap between the upper and lower bounds is huge.

While no progress has been made on the general form of the problem, some progress has been made in interesting special cases. Specifically, if the bounded convex loss functions are also assumed to be Lipschitz, Flaxman et al. [11] improves their regret guarantee to $\mathcal{R}(T) = \widetilde{O}(T^{3/4})$. If the loss functions are smooth (namely, their gradients are Lipschitz), Saha and Tewari [15] present an algorithm with a guaranteed regret of $\widetilde{O}(T^{2/3})$. Similarly, if the loss functions are bounded, Lipschitz, and strongly convex, the guaranteed regret is $\widetilde{O}(T^{2/3})$ [3]. If even stronger assumptions are made, an optimal regret rate of $\widetilde{\Theta}(T^{1/2})$ can be guaranteed; namely, when the loss functions are both smooth

and strongly-convex [12], when they are Lipschitz and linear [2], and when Lipschitz loss functions are not generated adversarially but drawn i.i.d. from a fixed and unknown distribution [4].

Recently, Bubeck et al. [8] made progress that did not rely on additional assumptions, such as Lipschitz, smoothness, or strong convexity, but instead considered the general problem in the one-dimensional case. That result proves that there exists an algorithm with optimal $\widetilde{\Theta}(T^{1/2})$ regret for arbitrary univariate convex functions $f_t : [0, 1] \mapsto [0, 1]$. Subsequently, and after the current paper was written, Bubeck and Eldan [7] generalized this result to bandit convex optimization in general Euclidean spaces (albeit requiring a Lipschitz assumption). However, the proofs in both papers are non-constructive and do not give any hint on how to construct a concrete algorithm, nor any indication that an efficient algorithm exists.

The current state of the bandit convex optimization problem has given rise to two competing conjectures. Some believe that there exists an efficient algorithm that matches the current lower bound. Meanwhile, others are trying to prove larger lower bounds, in the spirit of [10], even under the assumption that the loss functions are smooth; if the $\Omega(T^{1/2})$ lower bound is loose, a natural guess of the true regret rate would be $\widetilde{\Theta}(T^{2/3})$.[1] In this paper, we take an important step towards the resolution of this problem by presenting an algorithm that guarantees a regret of $\widetilde{\Theta}(T^{5/8})$ against any sequence of bounded, convex, smooth loss functions. Compare this result to the previous state-of-the-art result of $\widetilde{\Theta}(T^{2/3})$ (noting that $2/3 = 0.666...$ and $5/8 = 0.625$). This result rules out the possibility of proving a lower bound of $\Omega(T^{2/3})$ with smooth functions. While there remains a sizable gap with the $T^{1/2}$ lower bound, our result brings us closer to finding the elusive optimal algorithm for bandit convex optimization, at least in the case of smooth functions.

Our algorithm is a variation on the algorithms presented in [11, 1, 15], with one new idea. These algorithms all follow the same template: on each round, the algorithm computes an estimate of $\nabla f_t(x_t)$, the gradient of the current loss function at the current point, by applying a random perturbation to $x_t$. The sequence of gradient estimates is then plugged into a first-order online optimization technique. The technical challenge in the analysis of these algorithms is to bound the bias and the variance of these gradient estimates. Our idea is take a window of consecutive gradient estimates and average them, producing a new gradient estimate with lower variance and higher bias. Overall, the new bias-variance tradeoff works in our favor and allows us to improve the regret upper-bound.

Averaging uncorrelated random vectors to reduce variance is a well-known technique, but applying it in the context of bandit convex optimization algorithm is easier said than done and requires us to overcome a number of technical difficulties. For example, the gradient estimates in our window are taken at different points, which introduces a new type of bias. Another example is the difficulty that arrises when the sequence $x_s, \ldots, x_t$ travels adjacent to the boundary of the convex set $K$ (imagine transitioning from one face of a hypercube to another); the random perturbation applied to $x_s$ and $x_t$ could be supported on orthogonal directions, yet we average the resulting gradient estimates and expect to get a meaningful low-variance gradient estimate. While the basic idea is simple, our non-trivial technical analysis is not, and may be of independent interest.

## 2 Preliminaries

We begin by defining smooth bandit convex optimization more formally, and recalling several basic results from previous work on the problem (Flaxman et al. [11], Abernethy et al. [2], Saha and Tewari [15]) that we use in our analysis. We also review the necessary background on self-concordant barrier functions.

### 2.1 Smooth Bandit Convex Optimization

In the bandit convex optimization problem, an adversary first chooses a sequence of convex functions $f_1, \ldots, f_T : K \mapsto [0, 1]$, where $K$ is a closed and convex domain in $\mathbb{R}^d$. Then, on each round $t = 1, \ldots, T$, a randomized decision maker has to choose a point $x_t \in K$, and after committing to his decision he incurs a loss of $f_t(x_t)$, and observes this loss as feedback. The decision maker's expected loss (where expectation is taken with respect to his random choices) is $\mathbb{E}[\sum_{t=1}^{T} f_t(x_t)]$ and

his *regret* is

$$\mathcal{R}(T) = \mathbb{E}\left[\sum_{t=1}^{T} f_t(x_t)\right] - \min_{x \in K} \sum_{t=1}^{T} f_t(x) \ .$$

Throughout, we use the notation $\mathbb{E}_t[\cdot]$ to indicate expectations conditioned on all randomness up to and including round $t-1$.

We make the following assumptions. First, we assume that each of the functions $f_1, \ldots, f_T$ is $L$-Lipschitz with respect to the Euclidean norm $\|\cdot\|_2$, namely that $|f_t(x) - f_t(y)| \leq L\|x - y\|_2$ for all $x, y \in K$. We further assume that $f_t$ is $H$-smooth with respect to $\|\cdot\|_2$, which is to say that

$$\forall\, x, y \in K\,, \qquad \|\nabla f_t(x) - \nabla f_t(y)\|_2 \ \leq\ H\,\|x - y\|_2 \ .$$

In particular, this implies that $f_t$ is continuously differentiable over $K$. Finally, we assume that the Euclidean diameter of the decision domain $K$ is bounded by $D > 0$.

## 2.2 First Order Algorithms with Estimated Gradients

The online convex optimization problem becomes much easier in the *full information* setting, where the decision maker's feedback includes the vector $g_t = \nabla f_t(x_t)$, the gradient (or subgradient) of $f_t$ at the point $x_t$. In this setting, the decision maker can use a first-order online algorithm, such as the *projected online gradient descent* algorithm [17] or *dual averaging* [13] (sometimes known as *follow the regularized leader* [16]), and guarantee a regret of $O(T^{1/2})$. The dual averaging approach sets $x_t$ to be the solution to the following optimization problem,

$$x_t = \underset{x \in K}{\arg\min}\left\{ x \cdot \sum_{s=1}^{t-1} \alpha_{s,t} g_s + R(x) \right\} \ , \tag{1}$$

where $R$ is a suitably chosen regularizer, and for all $t = 1, \ldots, T$ and $s = 1, \ldots, t$ we define a non-negative weight $\alpha_{s,t}$. Typically, all of the weights $(\alpha_{s,t})$ are set to a constant value $\eta$, called the *learning rate parameter*.

However, since we are not in the full information setting and the decision maker does not observe $g_t$, the algorithms mentioned above cannot be used directly. The key observation of Flaxman et al. [11], which is later reused in all of the follow-up work, is that $g_t$ can be estimated by randomly perturbing the point $x_t$. Specifically, on round $t$, the algorithm chooses the point

$$y_t = x_t + \delta A_t u_t \ , \tag{2}$$

instead of the original point $x_t$, where $\delta > 0$ is a parameter that controls the magnitude of the perturbation, $A_t$ is a positive definite $d \times d$ matrix, and $u_t$ is drawn from the uniform distribution on the unit sphere. In Flaxman et al. [11], $A_t$ is simply set to the identity matrix whereas in Saha and Tewari [15], $A_t$ is more carefully tailored to the point $x_t$ (see details below). In any case, care should be taken to ensure that the perturbed point $y_t$ remains in the convex set $K$. The observed value $f_t(y_t)$ is then used to compute the gradient estimate

$$\hat{g}_t = \frac{d}{\delta} f_t(y_t) A_t^{-1} u_t \ , \tag{3}$$

and this estimate is fed to the first-order optimization algorithm. While $\hat{g}_t$ is not an unbiased estimator of $\nabla f_t(x_t)$, it is an unbiased estimator for the gradient of a different function, $\hat{f}_t$, defined by

$$\hat{f}_t(x) = \mathbb{E}_t[f_t(x + \delta A_t v)] \ , \tag{4}$$

where $v \in \mathbb{R}^d$ is uniformly drawn from the unit ball. The function $\hat{f}_t(x)$ is a smoothed version of $f_t$, which plays a key role in our analysis and in many of the previous results on this topic. The main property of $\hat{f}_t$ is summarized in the following lemma.

**Lemma 1** (Flaxman et al. [11], Saha and Tewari [15, Lemma 5]). *For any differentiable function $f : \mathbb{R}^d \mapsto \mathbb{R}$, positive definite matrix $A$, $x \in \mathbb{R}^d$, and $\delta \in (0, 1]$, define $\hat{g} = (d/\delta)f(x + \delta Au) \cdot A^{-1}u$, where $u$ is uniform on the unit sphere. Also, let $\hat{f}(x) = \mathbb{E}[f(x + \delta Av)]$ where $v$ is uniform on the unit ball. Then $\mathbb{E}[\hat{g}] = \nabla \hat{f}(x)$.*

The difference between $\nabla f_t(x_t)$ and $\nabla \hat{f}_t(x_t)$ is the bias of the gradient estimator $\hat{g}_t$. The analysis in Flaxman et al. [11], Abernethy et al. [2], Saha and Tewari [15] focuses on bounding the bias and the variance of $\hat{g}_t$ and their effect on the first-order optimization algorithm.

## 2.3 Self-Concordant Barriers

Following [2, 1, 15], our algorithm and analysis rely on the properties of self-concordant barrier functions. Intuitively, a barrier is a function defined on the interior of the convex body $K$, which is rather flat in most of the interior of $K$ and explodes to $\infty$ as we approach its boundary. Additionally, a self-concordant barrier has some technical properties that are useful in our setting. Before giving the formal definition of a self-concordant barrier, we define the local norm defined by a self-concordant barrier.

**Definition 2** (Local Norm Induced by a Self-Concordant Barrier [14]). Let $R : \operatorname{int}(K) \mapsto \mathbb{R}$ be a self-concordant barrier. The local norm induced by $R$ at the point $x \in \operatorname{int}(K)$ is denoted by $\|z\|_x$ and defined as $\|z\|_x = \sqrt{z^\mathsf{T} \nabla^2 R(x) z}$. Its dual norm is $\|z\|_{x,*} = \sqrt{z^\mathsf{T} (\nabla^2 R(x))^{-1} z}$.

In words, the local norm at $x$ is the Mahalanobis norm defined by the Hessian of $R$ at the point $x$, namely, $\nabla^2 R(x)$. We now give a formal definition of a self-concordant barrier.

**Definition 3** (Self-Concordant Barrier [14]). Let $K \subseteq \mathbb{R}^d$ be a convex body. A function $R : \operatorname{int}(K) \mapsto \mathbb{R}$ is a $\vartheta$-self-concordant barrier for $K$ if (i) $R$ is three times continuously differentiable, (ii) $R(x) \to \infty$ as $x \to \partial K$, and (iii) for all $x \in \operatorname{int}(K)$ and $y \in \mathbb{R}^d$, $R$ satisfies

$$|\nabla^3 R(x)[y, y, y]| \le 2\|y\|_x^3 \quad \text{and} \quad |\nabla R(x) \cdot y| \le \sqrt{\vartheta}\|y\|_x .$$

This definition is given for completeness, and is not directly used in our analysis. Instead, we rely on some useful properties of self-concordant barriers. First and foremost, there exists a $O(d)$-self-concordant barrier for any convex body [14, 6]. Efficiently-computable self-concordant barriers are only known for specific classes of convex bodies, such as polytopes, yet we make the standard assumption that we have an efficiently computable $\vartheta$-self-concordant barrier for the set $K$.

Another key feature of a self-concordant barrier is the set of *Dikin ellipsoids* that it defines. The Dikin ellipsoid at $x \in \operatorname{int}(K)$ is simply the unit ball with respect to the local norm at $x$. A key feature of the Dikin ellipsoid is that it is entirely contained in the convex body $K$, for any $x$ [see 14, Theorem 2.1.1]. Another technical property of a self-concordant barriers is that its Hessian changes slowly with respect to its local norm.

**Theorem 4** (Nesterov and Nemirovskii [14, Theorem 2.1.1]). *Let $K$ be a convex body with self-concordant barrier $R$. For any $x \in \operatorname{int}(K)$ and $z \in \mathbb{R}^d$ such that $\|z\|_x < 1$, it holds that*

$$(1 - \|z\|_x)^2 \, \nabla^2 R(x) \ \preceq \ \nabla^2 R(x + z) \ \preceq \ (1 - \|z\|_x)^{-2} \, \nabla^2 R(x) .$$

While the self-concordant barrier explodes to infinity at the boundary of $K$, it is quite flat at points that are far from the boundary. To make this statement formal, we define an operation that multiplicatively shrinks the set $K$ toward the minimizer of $R$ (called the analytic center of $K$). Let $y = \arg\min R(x)$ and assume without loss of generality that $R(y) = 0$. For any $\epsilon \in (0, 1)$ let $K_{y,\epsilon}$ denote the set $\{y + (1 - \epsilon)(x - y) : x \in K\}$. The next theorem states that the barrier is flat in $K_{y,\epsilon}$ and explodes to $\infty$ in the thin shell between $K_{y,\epsilon}$ and $K$.

**Theorem 5** (Nesterov and Nemirovskii [14, Propositions 2.3.2-3]). *Let $K$ be a convex body with $\vartheta$-self-concordant barrier $R$, let $y = \arg\min R(x)$, and assume that $R(y) = 0$. For any $\epsilon \in (0, 1]$, it holds that*

$$\forall \, x \in K_{y,\epsilon} \qquad R(x) \le \vartheta \log \frac{1}{\epsilon} .$$

Our assumptions on the loss functions, as the Lipschitz assumption or the smoothness assumption, are stated in terms of the standard Euclidean norm (which we denote by $\|\cdot\|_2$). Therefore, we will need to relate the Euclidean norm to the local norms defined by the self-concordant barrier. This is accomplished by the following lemma (whose proof appears in the supplementary material).

**Lemma 6.** *Let $K$ be a convex body with self-concordant barrier $R$ and let $D$ be the (Euclidean) diameter of $K$. For any $x \in K$, it holds that $D^{-1}\|z\|_{x,*} \le \|z\|_2 \le D\|z\|_x$ for all $z \in \mathbb{R}^d$.*

## 2.4 Self-Concordant Barrier as a Regularizer

Looking back at the dual averaging strategy defined in Eq. (1), we can now fill in some of the details that were left unspecified: [1, 15] set the regularization $R$ in Eq. (1) to be a $\vartheta$-self-concordant barrier for the set $K$. We use the following useful lemma from Abernethy and Rakhlin [1] in our analysis.

---
**Algorithm 1:** Bandit Smooth Convex Optimization

---
**Parameters:** perturbation parameter $\delta \in (0, 1]$, dual averaging weights $(\alpha_{s,t})$, self-concordant barrier $R : \text{int}(K) \mapsto \mathbb{R}$
**Initialize:** $y_1 \in K$ arbitrarily

**for** $t = 1, \ldots, T$
$\quad A_t \leftarrow (\nabla^2 R(x_t))^{-1/2}$
$\quad$ draw $u_t$ uniformly from the unit sphere
$\quad y_t \leftarrow x_t + \delta A_t u_t$
$\quad$ choose $y_t$, receive feedback $f_t(y_t)$
$\quad \hat{g}_t \leftarrow (d/\delta) f_t(y_t) \cdot A_t^{-1} u_t$
$\quad x_{t+1} \leftarrow \arg\min_{x \in K} \{x \cdot \sum_{s=1}^{t} \alpha_{s,t} \hat{g}_s + R(x)\}$

---

**Lemma 7** (Abernethy and Rakhlin [1]). *Let $K$ be a convex body with $\vartheta$-self-concordant barrier $R$, let $g_1, \ldots, g_T$ be vectors in $\mathbb{R}^d$, and let $\eta > 0$ be such that $\eta \|g_t\|_{x_t, *} \leq \frac{1}{4}$ for all $t$. Define $x_t = \arg\min_{x \in K} \{x \cdot \sum_{s=1}^{t-1} \eta g_s + R(x)\}$. Then,*

*(i) for all $t$ it holds that $\|x_t - x_{t+1}\|_{x_t} \leq 2\eta \|g_t\|_{x_t, *}$;*
*(ii) for any $x^\star \in K$ it holds that $\sum_{t=1}^{T} g_t \cdot (x_t - x^\star) \leq \frac{1}{\eta} R(x^\star) + 2\eta \sum_{t=1}^{T} \|g_t\|_{x_t, *}^2$.*

Algorithms for bandit convex optimization that use a self-concordant regularizer also use the same self-concordant barrier to obtain gradient estimates. Namely, these algorithms perturb the dual averaging solution $x_t$ as in Eq. (3), with the perturbation matrix $A_t$ set to $(\nabla^2 R(x_t))^{-1/2}$, the root of the inverse Hessian of $R$ at the point $x_t$. In other words, the distribution of $y_t$ is supported on the Dikin ellipsoid centered at $x_t$, scaled by $\delta$. Since $\delta \in (0, 1]$, this form of perturbation guarantees that $y_t \in K$. Moreover, if $y_t$ is generated in this way and used to construct the gradient estimator $\hat{g}_t$, then the local norm of $\hat{g}_t$ is bounded, as specified in the following lemma.

**Lemma 8** (Saha and Tewari [15, Lemma 5]). *Let $K \subseteq \mathbb{R}^d$ be a convex body with self-concordant barrier $R$. For any differentiable function $f : K \mapsto [0, 1]$, $\delta \in (0, 1]$, and $x \in \text{int}(K)$, define $\hat{g} = (d/\delta) f(y) \cdot A^{-1} u$, where $A = (\nabla^2 R(x))^{-1/2}$, $y = x + \delta A u$, and $u$ is drawn uniformly from the unit sphere. Then $\|\hat{g}\|_x \leq d/\delta$.*

## 3   Main Result

Our algorithm for the bandit smooth convex optimization problem is a variant of the algorithm in Saha and Tewari [15], and appears in Algorithm 1. Following Abernethy and Rakhlin [1], Saha and Tewari [15], we use a self-concordant function as the dual averaging regularizer and we use its Dikin ellipsoids to perturb the points $x_t$. The difference between our algorithm and previous ones is the introduction of dual averaging weights $(\alpha_{s,t})$, for $t = 1, \ldots, T$ and $s = 1, \ldots, t$, which allow us to vary the weight of each gradient in the dual averaging objective function.

In addition to the parameters $\delta$, $\eta$, and $\epsilon$, we introduce a new *buffering parameter* $k$, which takes non-negative integer values. We set the dual averaging weights in Algorithm 1 to be

$$\alpha_{s,t} = \begin{cases} \eta & \text{if } s \leq t - k \\ \frac{t-s+1}{k+1} \eta & \text{if } s > t - k \end{cases}, \tag{5}$$

where $\eta > 0$ is a global learning rate parameter. This choice of $(\alpha_{s,t})$ effectively decreases the influence of the feedback received on the most recent $k$ rounds. If $k = 0$, all of the $(\alpha_{s,t})$ become equal to $\eta$ and Algorithm 1 reduces to the algorithm in Saha and Tewari [15]. The surprising result is that there exists a different setting of $k > 0$ that gives a better regret bound.

We introduce a slight abuse of notation, which helps us simplify the presentation of our regret bound. We will eventually achieve the desired regret bound by setting the parameters $\eta$, $\delta$, and $k$ to be some functions of $T$. Therefore, from now on, we treat the notation $\eta$, $\delta$, and $k$ as an abbreviation for the functional forms $\eta(T)$, $\delta(T)$, and $k(T)$ respectively. The benefit is that we can now use asymptotic notation (e.g., $O(\eta k)$) to sweep meaningless low-order terms under the rug.

We prove the following regret bound for this algorithm.

**Theorem 9.** *Let $f_1, \ldots, f_T$ be a sequence of loss functions where each $f_t : K \mapsto [0,1]$ is differentiable, convex, $H$-smooth and $L$-Lipschitz, and where $K \subseteq \mathbb{R}^d$ is a convex body of diameter $D > 0$ with $\vartheta$-self-concordant barrier $R$. For any $\delta, \eta \in (0,1]$ and $k \in \{0, 1, \ldots, T\}$ assume that Algorithm 1 is run with these parameters and with the weights defined in Eq. (5) (using $k$ and $\eta$) to generate the sequences $x_1, \ldots, x_T$ and $y_1, \ldots, y_T$. If $12k\eta d \leq \delta$ and for any $\epsilon \in (0,1)$ it holds that*

$$\mathcal{R}(T) \;\leq\; HD^2\delta^2 T + \frac{\vartheta \log \frac{1}{\epsilon}}{\eta} + \frac{64d^2\eta T}{\delta^2(k+1)} + \frac{12(HD^2 + DL)d\eta\sqrt{k}T}{\delta} + O\!\left(\frac{T\epsilon}{\delta} + T\eta k\right) .$$

*Specifically, if we set $\delta = d^{1/4}T^{-3/16}$, $\eta = d^{-1/2}T^{-5/8}$, $k = d^{1/2}T^{1/8}$, and $\epsilon = T^{-100}$, we get that $\mathcal{R}(T) = O(\sqrt{d}\,T^{5/8}\log T)$.*

Note that if we set $k = 0$ in our theorem, we recover the $\widetilde{O}(T^{2/3})$ bound in Saha and Tewari [15] up to a small numerical constant (namely, the dependence on $L$, $H$, $D$, $\vartheta$, $d$, and $T$ is the same).

## 4   Analysis

Using the notation $x^\star = \arg\min_{x \in K} \sum_{t=1}^T f_t(x)$, the decision maker's regret becomes $\mathcal{R}(T) = \mathbb{E}\big[\sum_{t=1}^T f_t(y_t) - f_t(x^\star)\big]$. Following Flaxman et al. [11], Saha and Tewari [15], we rewrite the regret as

$$\mathcal{R}(T) = \underbrace{\mathbb{E}\left[\sum_{t=1}^T f_t(y_t) - \hat{f}_t(x_t)\right]}_{a} + \underbrace{\mathbb{E}\left[\sum_{t=1}^T \hat{f}_t(x_t) - \hat{f}_t(x^\star)\right]}_{b} + \underbrace{\mathbb{E}\left[\sum_{t=1}^T \hat{f}_t(x^\star) - f_t(x^\star)\right]}_{c} . \quad (6)$$

This decomposition essentially adds a layer of hallucination to the analysis: we pretend that the loss functions are $\hat{f}_1, \ldots, \hat{f}_T$ instead of $f_1, \ldots, f_T$ and we also pretend that we chose the points $x_1, \ldots, x_T$ rather than $y_1, \ldots, y_T$. We then analyze the regret in this pretend world (this regret is the expression in Eq. (6b)). Finally, we tie our analysis back to the real world by bounding the difference between that which we analyzed and the regret of the actual problem (this difference is the sum of Eq. (6a) and Eq. (6c)). The advantage of our pretend world over the real world is that we have unbiased gradient estimates $\hat{g}_1, \ldots, \hat{g}_T$ that can plug into the dual averaging algorithm.

The algorithm in Saha and Tewari [15] sets all of the dual averaging weights $(\alpha_{s,t})$ equal to the constant learning rate $\eta > 0$. It decomposes the regret as in Eq. (6) and their main technical result is the following bound for the individual terms:

**Theorem 10** (Saha and Tewari [15]). *Let $f_1, \ldots, f_T$ be a sequence of loss functions where each $f_t : K \mapsto [0,1]$ is differentiable, convex, $H$-smooth and $L$-Lipschitz, and where $K \subseteq \mathbb{R}^d$ is a convex body of diameter $D > 0$ and $\vartheta$-self-concordant barrier $R$. Assume that Algorithm 1 is run with perturbation parameter $\delta \in (0,1]$ and generates the sequences $x_1, \ldots, x_T$ and $y_1, \ldots, y_T$. Then for any $\epsilon \in (0,1)$ it holds that $(6a) + (6c) \leq (HD^2\delta^2 + \epsilon L)T$. If, additionally, the dual averaging weights $(\alpha_{s,t})$ are all set to the constant learning rate $\eta$ then $(6b) \leq \vartheta \log(1/\epsilon)\eta^{-1} + d^2\delta^{-2}\eta T$.*

The analysis in Saha and Tewari [15] goes on to obtain a regret bound of $\widetilde{O}(T^{2/3})$ by choosing optimal values for the parameters $\eta$, $\delta$ and $\epsilon$ and plugging those values into Theorem 10. Our analysis uses the first part of Theorem 10 to bound $(6a) + (6c)$ and shows that our careful choice of the dual averaging weights $(\alpha_{s,t})$ results in the following improved bound on $(6b)$.

We begin our analysis by defining a moving average of the functions $\hat{f}_1, \ldots, \hat{f}_T$, as follows:

$$\forall\, t = 1, \ldots, T, \qquad \bar{f}_t(x) \;=\; \frac{1}{k+1}\sum_{i=0}^k \hat{f}_{t-i}(x) , \qquad (7)$$

where, for soundness, we let $\bar{f}_s \equiv 0$ for $s \leq 0$. Also, define a moving average of gradient estimates:

$$\forall\, t = 1, \ldots, T, \qquad \bar{g}_t \;=\; \frac{1}{k+1}\sum_{i=0}^k \hat{g}_{t-i} ,$$

again, with $\hat{g}_s = 0$ for $s \le 0$. In Section 4 below, we show how each $\bar{g}_t$ can be used as a biased estimate of $\nabla \bar{f}_t(x_t)$. Also note that the choice of the dual averaging weights $(\alpha_{s,t})$ in Eq. (5) is such that $\sum_{s=1}^t \alpha_{s,t}\hat{g}_s = \eta \sum_{s=1}^t \bar{g}_s$ for all $t$. Therefore, the last step in Algorithm 1 basically performs dual averaging with the gradient estimates $\bar{g}_1, \dots, \bar{g}_T$ uniformly weighted by $\eta$.

We use the functions $\bar{f}_t$ to rewrite Eq. (6b) as

$$\underbrace{\mathbb{E}\left[\sum_{t=1}^T \hat{f}_t(x_t) - \bar{f}_t(x_t)\right]}_a + \underbrace{\mathbb{E}\left[\sum_{t=1}^T \bar{f}_t(x_t) - \bar{f}_t(x^\star)\right]}_b + \underbrace{\mathbb{E}\left[\sum_{t=1}^T \bar{f}_t(x^\star) - \hat{f}_t(x^\star)\right]}_c. \qquad (8)$$

This decomposition essentially adds yet another layer of hallucination to the analysis: we pretend that the loss functions are $\bar{f}_1, \dots, \bar{f}_T$ instead of $\hat{f}_1, \dots, \hat{f}_T$ (which are themselves pretend loss functions, as described above). Eq. (8b) is the regret in our new pretend scenario, while Eq. (8a) + Eq. (8c) is the difference between this regret and the regret in Eq. (6b).

The following lemma bounds each of the terms in Eq. (8) separately, and summarizes the main technical contribution of our paper.

**Lemma 11.** *Under the conditions of Theorem 9, for any $\epsilon \in (0, 1)$ it holds that (8c) $\le 0$,*

$$(8a) \le \frac{12DLd\eta\sqrt{k}T}{\delta} + O\big((1+\eta)k\big), \quad and$$

$$(8b) \le \frac{\vartheta \log \frac{1}{\epsilon}}{\eta} + \frac{64d^2\eta T}{\delta^2(k+1)} + \frac{12HD^2 d\eta\sqrt{k}T}{\delta} + O\left(\frac{T\epsilon}{\delta} + T\eta k\right).$$

## 4.1 Proof Sketch of Lemma 11

As mentioned above, the basic intuition of our technique is quite simple: average the gradients to decrease their variance. Yet, applying this idea in the analysis is tricky. We begin by describing the main source of difficulty in proving Lemma 11.

Recall that our strategy is to pretend that the loss functions are $\bar{f}_1, \dots, \bar{f}_T$ and to use the random vector $\bar{g}_t$ as a biased estimator of $\nabla \bar{f}_t(x_t)$. Naturally, one of our goals is to show that this bias is small. Recall that each $\hat{g}_s$ is an unbiased estimator of $\nabla \hat{f}_s(x_s)$ (conditioned on the history up to round $t$). Specifically, note that each vector in the sequence $\hat{g}_{t-k}, \dots, \hat{g}_t$ is a gradient estimate *at a different point*. Yet, we average these vectors and claim that they accurately estimate $\nabla \bar{f}_t$ at the current point, $x_t$. Luckily, $\hat{f}_t$ is $H$-smooth, so $\nabla \hat{f}_{t-i}(x_{t-i})$ should not be much different than $\nabla \hat{f}_{t-i}(x_t)$, provided that we show that $x_{t-i}$ and $x_t$ are close to each other in Euclidean distance.

To show that $x_{t-i}$ and $x_t$ are close, we exploit the stability of the dual averaging algorithm. Particularly, the first claim in Lemma 7 states that $\|x_s - x_{s+1}\|_{x_s}$ is controlled by $\|\bar{g}_s\|_{x_s,*}$ for all $s$, so now we need to show that $\|\bar{g}_s\|_{x_s,*}$ is small. However, $\bar{g}_t$ is the average of $k+1$ gradient estimates taken at different points; each $\hat{g}_{t-i}$ is designed to have a small norm with respect to its own local norm $\|\cdot\|_{x_{t-i},*}$; for all we know, it may be very large with respect to the current local norm $\|\cdot\|_{x_t,*}$. So now we need to show that the local norms at $x_{t-i}$ and $x_t$ are similar. We could prove this if we knew that $x_{t-i}$ and $x_t$ are close to each other—which is exactly what we set out to prove in the beginning. This chicken-and-egg situation complicates our analysis considerably.

Another non-trivial component of our proof is the variance reduction analysis. The motivation to average $\hat{g}_{t-k}, \dots, \hat{g}_t$ is to generate new gradient estimates with a smaller variance. While the random vectors $\hat{g}_1, \dots, \hat{g}_T$ are not independent, we show that their randomness is *uncorrelated*. Therefore, the variance of $\bar{g}_t$ is $k+1$ times smaller than the variance of each $\hat{g}_t$. However, to make this argument formal, we again require the local norms at $x_{t-i}$ and $x_t$ to be similar.

To make things more complicated, there is the recurring need to move back and forth between local norms and the Euclidean norm, since the latter is used in the definition of Lipschitz and smoothness. All of this has to do with bounding Eq. (8b), the regret with respect to the pretend loss functions $\bar{f}_1, \dots, \bar{f}_T$ - an additional bias term appears in the analysis of Eq. (8a).

We conclude the paper by stating our main lemmas and sketching the proof Lemma 11. The full technical proofs are all deferred to the supplementary material and replaced with some high level commentary.

To break the chicken-and-egg situation described above, we begin with a crude bound on $\|\bar{g}_t\|_{x_t,*}$, which does not benefit at all from the averaging operation. We simultaneously prove that the local norms at $x_{t-i}$ and $x_t$ are similar.

**Lemma 12.** *If the parameters $k$, $\eta$, and $\delta$ are chosen such that $12k\eta d \leq \delta$ then for all $t$,*

    *(i) $\|\bar{g}_t\|_{x_t,*} \leq 2d/\delta$;*
    *(ii) for any $0 \leq i \leq k$ such that $t - i \geq 1$ it holds that $\frac{1}{2}\|z\|_{x_{t-i},*} \leq \|z\|_{x_t,*} \leq 2\|z\|_{x_{t-i},*}$.*

Lemma 12 itself has a chicken-and-egg aspect, which we resolve using an inductive proof technique. Armed with the knowledge that the local norms at $x_{t-i}$ and $x_t$ are similar, we go on to prove the more refined bound on $\mathbb{E}_t\big[\|\bar{g}_t\|_{x_t,*}^2\big]$, which does benefit from averaging.

**Lemma 13.** *If the parameters $k$, $\eta$, and $\delta$ are chosen such that $12k\eta d \leq \delta$ then*

$$\mathbb{E}_t\big[\|\bar{g}_t\|_{x_t,*}^2\big] \ \leq \ 2D^2L^2 + \frac{32d^2}{\delta^2(k+1)} \ .$$

The proof constructs a martingale difference sequence and uses the fact that its increments are uncorrelated. Compare the above to Lemma 8, which proves that $\|\hat{g}_{t-i}\|_{x_{t-i},*}^2 \leq d^2/\delta^2$ and note the extra $k+1$ in our denominator—all of our hard work was aimed at getting this factor.

Next, we set out to bound the expected Euclidean distance between $x_{t-i}$ and $x_t$. This bound is later needed to exploit the $L$-Lipschitz and $H$-smooth assumptions. The crude bound on $\|\bar{g}_s\|_{x_s,*}$ from Lemma 12 is enough to satisfy the conditions of Lemma 7, which then tells us that $\mathbb{E}[\|x_s - x_{s+1}\|_{x_s}]$ is controlled by $\mathbb{E}[\|\bar{g}_s\|_{x_s,*}]$. The latter enjoys the improved bound due to Lemma 13. Integrating the resulting bound over time, we obtain the following lemma.

**Lemma 14.** *If the parameters $k$, $\eta$, and $\delta$ are chosen such that $12k\eta d \leq \delta$ then for all $t$ and any $0 \leq i \leq k$ such that $t - i \geq 1$, we have*

$$\mathbb{E}[\|x_{t-i} - x_t\|_2] \ \leq \ \frac{12Dd\eta\sqrt{k}}{\delta} \ + \ O(\eta k) \ .$$

Notice that $x_{t-i}$ and $x_t$ may be $k$ rounds apart, but the bound scales only with $\sqrt{k}$. Again, this is the work of the averaging technique.

Finally, we have all the tools in place to prove our main result, Lemma 11.

*Proof sketch.* The first term, Eq. (8a), is bounded by rewriting $\bar{f}_t(x_t) = \frac{1}{k+1}\sum_{i=0}^{k}\hat{f}_{t-i}(x_t)$ and then proving that $\hat{f}_{t-i}(x_t)$ is not very far from $\hat{f}_t(x_t)$. This follows from the fact that $\hat{f}_t$ is $L$-Lipschitz and from Lemma 14. To bound the second term, Eq. (8b), we use the convexity of each $\bar{f}_t$ to write

$$\mathbb{E}\left[\sum_{t=1}^{T}\bar{f}_t(x_t) - \bar{f}_t(x^\star)\right] \ \leq \ \mathbb{E}\left[\sum_{t=1}^{T}\nabla\bar{f}_t(x_t) \cdot (x_t - x^\star)\right] \ .$$

We relate the right-hand side above to

$$\mathbb{E}\left[\sum_{t=1}^{T}\bar{g}_t \cdot (x_t - x^\star)\right] \ ,$$

using the fact that $\hat{f}_t$ is $H$-smooth and again using Lemma 14. Then, we upper bound the above using Lemma 7, Theorem 5, and Lemma 13. $\qquad\square$

### Acknowledgments

We thank Jian Ding for several critical contributions during the early stages of this research. Parts of this work were done while the second and third authors were at Microsoft Research, the support of which is gratefully acknowledged.

## Footnotes

[1]In fact, we are aware of at least two separate research groups that invested time trying to prove such an $\Omega(T^{2/3})$ lower bound.

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
