[Supplementary Material]

# A  Proofs

## A.1  Proof of Lemma 6

*Proof.* Assume, without loss of generality, that $x$ is the origin of $\mathbb{R}^d$. Recall that the Dikin ellipsoid at $x$ is contained in $K$. By definition, $K$ is itself contained in a Euclidean ball of radius $D$ centered at $x$. Therefore the Dikin ellipsoid at $x$ is contained in the $D$-radius Euclidean ball. Therefore, the gauge function (Minkowski functional) defined by this Dikin ellipsoid (which is exactly $\| \cdot \|_x$) is greater or equal to the gauge function defined by the $D$-radius Euclidean ball (which is $D^{-1}\| \cdot \|_2$). We proved one direction of the bound in the claim:

$$\forall\, z \in \mathbb{R}^d \qquad \frac{1}{D}\|z\|_2^2 \;\le\; z^\mathsf{T}\nabla^2 R(x)z \;.$$

Plugging $z' = (\nabla^2 R(x))^{-1/2}z$ into the inequality above gives

$$\forall\, z' \in \mathbb{R}^d \qquad \frac{1}{D}z^\mathsf{T}(\nabla^2 R(x))^{-1}z \;\le\; \|z\|_2^2 \;,$$

which concludes the proof. $\qquad\qquad\qquad\qquad\qquad\qquad\qquad\qquad\qquad\qquad\qquad\qquad\quad\square$

## A.2  Proof of Lemma 12

*Proof.* We prove claim (i) by complete induction on $t$. For $t = 1$, it holds by definition that $\bar{g}_1 = \hat{g}_1/(k+1)$. Lemma 8 bounds $\|\hat{g}_s\|_{x_s,*} \le d/\delta$, and therefore $\|\bar{g}_1\|_{x_t,*} \le d/(\delta(k+1))$. The assumption that $k \ge 0$ proves the claim.

Next, we deal with $t > 1$. Using the triangle inequality, we have

$$\|\bar{g}_t\|_{x_t,*} \;\le\; \frac{1}{k+1}\sum_{i=0}^{k}\|\hat{g}_{t-i}\|_{x_t,*} \;. \tag{9}$$

Lemma 8 gives us an upper bound on $\|\hat{g}_{t-i}\|_{x_{t-i},*}$, which is not the same norm as in Eq. (9). Therefore, our proof strategy is to show that these two norms are only a factor of 2 apart.

Assume, by complete induction, that $\|\bar{g}_s\|_{x_s,*} \le 2d/\delta$ for all $s < t$. It follows that $\eta\|\bar{g}_s\|_{x_s,*} \le 2\eta d/\delta$. Using our assumption that $12k\eta d \le \delta$ and the fact that $k \ge 1$, we conclude that $\eta\|\bar{g}_s\|_{x_s,*} \le \frac{1}{4}$. Therefore, the condition of Lemma 7 holds and we have

$$\|x_{s+1} - x_s\|_{x_s} \;\le\; 2\eta\,\|\bar{g}_s\|_{x_s,*} \;.$$

Again using the induction hypothesis, we have

$$\|x_{s+1} - x_s\|_{x_s} \;\le\; \frac{4\eta d}{\delta} \;.$$

By our assumption that $12k\eta d \le \delta$ it follows that the right-hand side above is at most $1/3$, the conditions of Theorem 4 are satisfied, and we have

$$\left(1 - \tfrac{4\eta d}{\delta}\right)^2\left(\nabla^2 R(x_s)\right)^{-1} \;\preceq\; \left(\nabla^2 R(x_{s+1})\right)^{-1} \;\preceq\; \left(1 - \tfrac{4\eta d}{\delta}\right)^{-2}\left(\nabla^2 R(x_s)\right)^{-1} \;.$$

Recalling the definition of the dual local norm (see Definition 2), it follows that

$$\forall\, z \in \mathbb{R}^d\,, \quad \left(1 - \tfrac{4\eta d}{\delta}\right)\|z\|_{x_s,*} \;\le\; \|z\|_{x_{s+1},*} \;\le\; \left(1 - \tfrac{4\eta d}{\delta}\right)^{-1}\|z\|_{x_s,*} \;.$$

By applying this inequality recursively, we get that for any positive $s \in \{t-k+1, \ldots, t\}$

$$\forall\, z \in \mathbb{R}^d\,, \quad \left(1 - \tfrac{4\eta d}{\delta}\right)^k\|z\|_{x_s,*} \;\le\; \|z\|_{x_t,*} \;\le\; \left(1 - \tfrac{4\eta d}{\delta}\right)^{-k}\|z\|_{x_s,*} \;.$$

Next we show that $(1 - 4\eta d/\delta)^k$ is at least $\frac{1}{2}$. If $k = 0$, this is trivial. If $k > 0$ denote $\beta = 4\eta d/\delta$ and notice that $\beta \le \frac{1}{3}$ due to the assumption that $12k\eta d \le \delta$. Then, apply the inequality $1 - \beta \ge \exp(-2\beta)$, which is valid for any $\beta \in [0, \frac{1}{2}]$, to bound $(1 - 4\eta d/\delta)^k \ge \exp(-8k\eta d/\delta) \ge \exp(-\frac{2}{3}) \ge \frac{1}{2}$. Similarly, $(1 - 4\eta d/\delta)^{-k}$ is at most 2. We conclude that

$$\tfrac{1}{2}\|z\|_{x_s,*} \;\le\; \|z\|_{x_t,*} \;\le\; 2\|z\|_{x_s,*} \;, \tag{10}$$

which proves claim *(ii)*. Combining this bound with Eq. (9) gives

$$\|\bar{g}_t\|_{x_t,*} \leq \frac{2}{k+1} \sum_{i=0}^{k} \|\hat{g}_{t-i}\|_{x_{t-i},*} .$$

Applying the bound $\|\hat{g}_{t-i}\|_{x_{t-i},*} \leq d/\delta$ from Lemma 8 concludes the inductive proof. $\qquad\square$

We prove a simple corollary of Lemma 12, which shows that the assumption $12k\eta d \leq \delta$ (which we make throughout our analysis) implies the conditions of Lemma 7.

**Corollary 15.** *If $12k\eta d \leq \delta$ then $\eta\|\bar{g}_t\|_{x_t,*} \leq \frac{1}{4}$ for all t.*

*Proof.* Lemma 12 states that $\|\bar{g}_t\|_{x_t,*} \leq 2d/\delta$ and therefore $\eta\|\bar{g}_t\|_{x_t,*} \leq 2d\eta/\delta$. Since $k \geq 1$, the assumption that $12k\eta d \leq \delta$ implies that the above is at most $1/6$, and certainly less than $1/4$. $\qquad\square$

### A.3  $\hat{f}_t$ is *L*-Lipschitz and *H*-Smooth

An important property of the function $\hat{f}_t$ is that it retains the properties of $f_t$.

**Lemma 16.** *Let $f : K \mapsto [0,1]$ be differentiable L-Lipschitz, and H-smooth, and let $\hat{f}_t(x) = \mathbb{E}[f(x + \delta Av)]$, where $\delta > 0$, $A$ is full-rank $d \times d$ matrix and $v$ is a random vector. Then $\hat{f}$ is also L-Lipschitz and H-smooth.*

*Proof of Lemma 16.* First, we prove that $\hat{f}$ is *L*-Lipschitz. For any $x, y \in K$, we have

$$\hat{f}(x) - \hat{f}(y) = \mathbb{E}[f(x + \delta Av) - f(y + \delta Av)] \leq \mathbb{E}[L\|x - y\|_2] ,$$

where the inequality follows from the assumption that $f$ itself is *L*-Lipschitz.

Next, we prove that $\hat{f}$ is *H*-Smooth. For any $x, y \in K$, we have

$$\|\nabla\hat{f}(x) - \nabla\hat{f}(y)\| = \|\nabla\mathbb{E}[f(x + \delta Av)] - \nabla\mathbb{E}[f(y + \delta Av)]\| .$$

We can switch the order of the expectation and the differentiation due to uniform boundedness, and the right-hand side above becomes

$$\|\mathbb{E}[\nabla f(x + \delta Av) - \nabla f(y + \delta Av)]\| .$$

From Jensen's inequality, the above is bounded by

$$\mathbb{E}\Big[\|\nabla f(x + \delta Av) - \nabla f(y + \delta Av)\|\Big] .$$

The term inside the square brackets is bounded by $H\|x - y\|_2$ due to the assumption that $f$ is *H*-smooth. This proves that $\hat{f}$ is also *H*-smooth. $\qquad\square$

### A.4  Proof of Lemma 13

*Proof.* From the definition of $\bar{g}_t$ and the triangle inequality, we bound

$$\|\bar{g}_t\|_{x_t,*} = \frac{1}{k+1}\Big\|\sum_{i=0}^{k}\hat{g}_{t-i}\Big\|_{x_t,*} \leq \frac{1}{k+1}\Big\|\sum_{i=0}^{k}\mathbb{E}_{t-i}[\hat{g}_{t-i}]\Big\|_{x_t,*} + \frac{1}{k+1}\Big\|\sum_{i=0}^{k}\hat{g}_{t-i} - \mathbb{E}_{t-i}[\hat{g}_{t-i}]\Big\|_{x_t,*} .$$

Using $(\beta + \gamma)^2 \leq 2\beta^2 + 2\gamma^2$ and taking expectations, we have

$$\mathbb{E}[\|\bar{g}_t\|_{x_t,*}^2] \leq \frac{2}{k^2}\mathbb{E}\left[\Big\|\sum_{i=0}^{k}\mathbb{E}_{t-i}[\hat{g}_{t-i}]\Big\|_{x_t,*}^2\right] + \frac{2}{k^2}\mathbb{E}\left[\Big\|\sum_{i=0}^{k}\hat{g}_{t-i} - \mathbb{E}_{t-i}[\hat{g}_{t-i}]\Big\|_{x_t,*}^2\right] . \quad (11)$$

We deal separately with each of the two terms on the right-hand side above. Using Eq. (6), we have

$$\Big\|\sum_{i=0}^{k}\mathbb{E}_{t-i}[\hat{g}_{t-i}]\Big\|_{x_t,*} \leq D\Big\|\sum_{i=0}^{k}\mathbb{E}_{t-i}[\hat{g}_{t-i}]\Big\|_2 .$$

Triangle inequality upper bounds the right-hand side above by $D \sum_{i=0}^{k} \left\| \mathbb{E}_{t-i}[\hat{g}_{t-i}] \right\|_2$. By Theorem 1, this equals $D \sum_{i=0}^{k} \left\| \nabla \hat{f}_{t-i}(x_{t-i}) \right\|_2$. The functions $\hat{f}_{t-i}$ are $L$-Lipschitz due to Lemma 16, so the first term on the right-hand side of Eq. (11) is upper-bounded by $2D^2 L^2$.

Moving on to the second term in Eq. (11), we abbreviate $h_s = \hat{g}_s - \mathbb{E}_s[\hat{g}_s]$ for all $s$ and study the term $\mathbb{E}\left[ \left\| \sum_{i=0}^{k} h_{t-i} \right\|_{x_t,*}^2 \right]$. Using Lemma 12, we bound

$$ \mathbb{E}\left[ \left\| \sum_{i=0}^{k} h_{t-i} \right\|_{x_t,*}^2 \right] \ \leq \ 4\mathbb{E}\left[ \left\| \sum_{i=0}^{k} h_{t-i} \right\|_{x_{t-k},*}^2 \right] \ . $$

Using the definition of the local norm, we write the right-hand side above as

$$ 4\mathbb{E}\left[ \sum_{i=0}^{k} \sum_{j=0}^{k} h_{t-i}^\mathsf{T} \left( \nabla^2 R(x_{t-k}) \right)^{-1} h_{t-j} \right] \ . \tag{12} $$

Now note that the sequence $h_1, \ldots, h_T$ is a martingale difference sequence, and therefore its increments are uncorrelated. Namely, for $i > j$ it holds that

$$ \mathbb{E}_{t-j}\left[ h_{t-i}^\mathsf{T} \left( \nabla^2 R(x_{t-k}) \right)^{-1} h_{t-j} \right] \ = \ 0 \ . $$

Therefore, Eq. (12) equals

$$ 4 \sum_{i=0}^{k} \mathbb{E}\left[ \|h_{t-i}\|_{x_{t-k},*}^2 \right] \ . $$

Another application of Lemma 12 gives the bound

$$ \mathbb{E}\left[ \left\| \sum_{i=0}^{k} h_{t-i} \right\|_{x_t,*}^2 \right] \ \leq \ 16 \sum_{i=0}^{k} \mathbb{E}\left[ \|h_{t-i}\|_{x_{t-i},*}^2 \right] \ . \tag{13} $$

Each term $\mathbb{E}_s\left[ \|h_s\|_{x_s,*}^2 \right]$ satisfies

$$ \mathbb{E}_s\left[ h_s^\mathsf{T} \left( \nabla^2 R(x_s) \right)^{-1} h_s \right] \ = \ \mathbb{E}_s\left[ \hat{g}_s^\mathsf{T} \left( \nabla^2 R(x_s) \right)^{-1} \hat{g}_s \right] \ - \ \mathbb{E}_s[\hat{g}_s]^\mathsf{T} \left( \nabla^2 R(x_s) \right)^{-1} \mathbb{E}[\hat{g}_s] \ . $$

The right-most term above is non-negative because $\left( \nabla^2 R(x_s) \right)^{-1}$ is positive semi-definite, so $\mathbb{E}_s[\|h_s\|_{x_s,*}^2] \leq \mathbb{E}_s[\|\hat{g}_s\|_{x_s,*}^2]$, which Lemma 8 further bounds by $d^2/\delta^2$. Plugging this back into Eq. (13) gives

$$ \mathbb{E}\left[ \left\| \sum_{i=0}^{k} h_{t-i} \right\|_{x_t,*}^2 \right] \ \leq \ \frac{16kd^2}{\delta^2} \ . $$

Overall, we have shown that the second term on the right-hand side of Eq. (11) is upper-bounded by $32d^2/(\delta^2(k+1))$. □

## A.5  Proof of Lemma 14

*Proof.* Using triangle inequality, we get

$$ \mathbb{E}[\|x_{t-i} - x_t\|_2] \ \leq \ \sum_{s=t-i}^{t-1} \mathbb{E}[\|x_s - x_{s+1}\|_2] \ , $$

and we bound each term individually. For any $s$, using Lemma 6, it holds that

$$ \|x_s - x_{s+1}\|_2 \ \leq \ D\|x_s - x_{s+1}\|_{x_s} \ . \tag{14} $$

By Corollary 15, the conditions of Lemma 7 are met, so the right-hand side of Eq. (14) is bounded by $2D\eta\|\bar{g}_s\|_{x_s,*}$. Therefore

$$ \mathbb{E}\left[ \|x_s - x_{s+1}\|_2 \right] \ \leq \ 2D\eta \, \mathbb{E}\left[ \|\bar{g}_s\|_{x_s,*} \right] \ . \tag{15} $$

Using Jensen's inequality, we have

$$\mathbb{E}\big[\|\bar{g}_s\|_{x_s,*}\big] \;\leq\; \sqrt{\mathbb{E}\big[\|\bar{g}_s\|_{x_s,*}^2\big]}$$

The right-hand side above is upper-bounded by Lemma 13. Using the fact that $\sqrt{a+b} \leq \sqrt{a} + \sqrt{b}$, we obtain

$$\mathbb{E}\big[\|\bar{g}_s\|_{x_s,*}\big] \;\leq\; 2D^2 L^2 + \frac{6d}{\delta\sqrt{k+1}} \; .$$

Plugging back into Eq. (15) gives

$$\mathbb{E}\big[\|x_s - x_{s+1}\|_2\big] \;\leq\; 4D^3 L^2 \eta + \frac{12 D d \eta}{\delta\sqrt{k}} \; .$$

The claim follows from the assumption that $i \leq k$. $\qquad\square$

## A.6   Proof of Lemma 11

We begin with a technical lemma.

**Lemma 17.** *Assume that the parameters $k$, $\eta$, and $\delta$ are chosen such that $12k\eta d \leq \delta$, and for any $\epsilon \in (0,1)$ it holds that*

$$\sum_{t=1}^{T} \bar{g}_t \cdot (x_t - x^\star) \;\leq\; \frac{\vartheta \log \frac{1}{\epsilon}}{\eta} + 2\eta \sum_{t=1}^{T} \|\bar{g}_t\|_{x_t,*}^2 \;+\; O(\epsilon T/\delta).$$

*Proof.* Let $y$ be the analytic center of the convex body $K$ and recall the definition of the shrunk set $K_{y,\epsilon}$ in Section 2.3. Define $x' = \arg\min_{x \in K_{y,\epsilon}} \|x - x^\star\|_2$, the Euclidean projection of $x^\star$ onto $K_{y,\epsilon}$. Since $x' \in K_{y,\epsilon}$, Theorem 5 states that

$$R(x) \;\leq\; \vartheta \log \frac{1}{\epsilon} \; .$$

Lemma 7 holds due to Corollary 15, and states that

$$\sum_{t=1}^{T} \bar{g}_t \cdot (x_t - x') \;\leq\; \frac{1}{\eta}\vartheta \log \frac{1}{\epsilon} + 2\eta \sum_{t=1}^{T} \|g_t\|_{x_t,*}^2 \; .$$

To replace $x'$ with $x^\star$ in the above, we note that

$$\sum_{t=1}^{T} \bar{g}_t \cdot (x_t - x^\star) \;-\; \sum_{t=1}^{T} \bar{g}_t \cdot (x_t - x') \;\leq\; T \|\bar{g}_t\|_2 \, \|x' - x^\star\|_2 \; .$$

Since the diameter of $K$ is $D$, it follows that $\|x' - x^\star\|_2 \leq \epsilon D$. The norm $\|\bar{g}_t\|_2$ is bounded by $D\|\bar{g}_t\|_{x_t,*}$, Lemma 12 bounds by $2dD/\delta$. $\qquad\square$

We are ready to prove the main theorem.

*Proof of Lemma 11.* By the definition of $\bar{f}_t$ in Eq. (7), it holds that

$$\sum_{t=1}^{T} \hat{f}_t(x_t) - \bar{f}_t(x_t) \;=\; \frac{1}{k+1} \sum_{t=1}^{T-k} \sum_{i=1}^{k} \hat{f}_t(x_{t+i}) - \hat{f}_t(x_t) \;+\; \sum_{t=T-k+1}^{T} \frac{T-t+1}{k+1} \hat{f}_t(x_t) \; .$$

Lemma 16 tells us that each $\hat{f}_t$ is $L$-Lipschitz, and we know that its range is $[0,1]$. Therefore, the above can be upper-bounded by

$$\frac{1}{k+1} \sum_{t=1}^{T-k} \sum_{i=1}^{k} L\|x_{t+i} - x_t\|_2 \;+\; k \; .$$

Lemma 14 bounds each term $\|x_{t+i} - x_t\|_2$ and establishes the stated bound over Eq. (8a).

Moving on to Eq. (8b), we use the convexity of $\bar{f}$ to bound

$$\mathbb{E}\left[\sum_{t=1}^{T}\bar{f}_t(x_t) - \bar{f}_t(x^\star)\right] \leq \mathbb{E}\left[\sum_{t=1}^{T}\nabla\bar{f}_t(x_t)\cdot(x_t - x^\star)\right]$$

$$= \mathbb{E}\left[\sum_{t=1}^{T}\left(\nabla\bar{f}_t(x_t) - \bar{g}_t\right)\cdot(x_t - x^\star)\right] + \mathbb{E}\left[\sum_{t=1}^{T}\bar{g}_t\cdot(x_t - x^\star)\right] \ .$$

Recalling that $x_t$ is the result of a dual averaging step using the gradient estimates $\bar{g}_1,\ldots,\bar{g}_{t-1}$, Lemma 17 states that

$$\mathbb{E}\left[\sum_{t=1}^{T}\bar{g}_t\cdot(x_t - x^\star)\right] \leq \frac{\vartheta\log\frac{1}{\epsilon}}{\eta} + 2\eta\sum_{t=1}^{T}\mathbb{E}\left[\|\bar{g}_t\|_{x_t,*}^2\right] + O(\epsilon T/\delta) \ .$$

Using Lemma 13, we bound the above by

$$\frac{\vartheta\log\frac{1}{\epsilon}}{\eta} + \frac{64d^2\eta T}{\delta^2(k+1)} + O\big(T(\epsilon/\delta + \eta)\big) \ .$$

It remains to bound the term

$$\mathbb{E}\left[\sum_{t=1}^{T}\left(\nabla\bar{f}_t(x_t) - \bar{g}_t\right)\cdot(x_t - x^\star)\right] \ . \tag{16}$$

Using the definitions of $\bar{f}_t$ and $\bar{g}_t$, we rewrite it as

$$\frac{1}{k+1}\sum_{t=1}^{T}\sum_{i=0}^{k}\mathbb{E}\left[\left(\nabla\hat{f}_{t-i}(x_t) - \hat{g}_{t-i}\right)\cdot(x_t - x^\star)\right] \ .$$

Note that the random variable $x_s$ is determined by the randomness before time $s$, while $\hat{g}_s$ is only determined when we expose the randomness on round $s$. Therefore, This term equals

$$\frac{1}{k+1}\sum_{t=1}^{T}\sum_{i=0}^{k}\mathbb{E}\left[\left(\nabla\hat{f}_{t-i}(x_t) - \mathbb{E}_{t-i}[\hat{g}_{t-i}]\right)\cdot(x_t - x^\star)\right] \ .$$

Using Lemma 1 states that $\mathbb{E}_{t-i}[\hat{g}_{t-i}]$ above equals $\nabla\hat{f}_{t-i}(x_{t-i})$. Cauchy-Schwartz bounds this by

$$\frac{1}{k+1}\sum_{t=1}^{T}\sum_{i=0}^{k}\mathbb{E}\left[\|\nabla\hat{f}_{t-i}(x_{t-i}) - \nabla\hat{f}_{t-1}(x_{t-i})\|_2\,\|x_t - x^\star\|_2\right] \ .$$

The term $\|x_t - x^\star\|_2$ is bounded by $D$. Lemma 16 states that $\hat{f}_{t-i}$ is $H$-smooth; together with Lemma 14 we have the bound

$$\|\nabla\hat{f}_{t-i}(x_{t-i}) - \nabla\hat{f}_{t-1}(x_{t-i})\|_2 \leq 12HDd\frac{\eta\sqrt{k}}{\delta} + O(\eta k) \ .$$

Plugging these bounds back into Eq. (16) gives the desired bound of Eq. (8b).

Moving on to the last term,

$$\mathbb{E}\left[\sum_{t=1}^{T}\bar{f}_t(x^\star) - \hat{f}_t(x^\star)\right] \ ,$$

it is not hard to show that the sum inside the expectation is non-positive. Indeed, each $\bar{f}_t$ is a moving average of the $\hat{f}_t$'s, so that up to boundary effects the sum of the $\bar{f}_t(x^\star)$ terms equals the sum of the $\hat{f}_t(x^\star)$ ones; considering the boundary effects, we see that the first sum can only be smaller than the latter. This implies that Eq. (8c) $\leq 0$, and concludes the proof. $\qquad\square$