[Reviews · NeurIPS 2015]

Submitted by Assigned_Reviewer_1

The paper studies bandit convex optimization problem in the adversarial setting. Loss functions are assumed to be Lipschitz and smooth. The state-of-the-art result for this setting is algorithm of Saha and Tewari with regret bound of O(T^{2/3}). Main contribution of the current paper is an algorithm that enjoys an improved bound of O(T^{5/8}). Similar to the earlier work, the algorithm is based on a reduction to online convex optimization. To achieve the improved bound, the sequence of gradient estimates are weighted non-uniformly such that the influence of the most recent estimates is decreased.

I enjoyed reading this paper. It addresses an important question and the proposed algorithm is simple and intuitive. The paper is very well-written and provides an excellent overview of existing techniques and the main ideas.
Summary: I enjoyed reading this paper. It addresses an important question and the proposed algorithm is simple and intuitive. The paper is very well-written and provides an excellent overview of existing techniques and the main ideas.

Submitted by Assigned_Reviewer_2

As I have stated in review summary, the significance of the paper, as least in my opinion, cannot be questioned. It is a simple question of whether the proofs are correct. I will highlight a couple of points which I believe are mistakes. Other than that, I have a few minor questions regarding presentation of the paper (which are not important in terms of accept/reject decision, but if the paper gets in, will be important in terms of clarity).

Line 322-323- The authors state that by definitions of \gar{g}_t and the \alpha variables, it is true that \sum_{s=1}^t \alpha_{s,t} \hat{g}_s= \sum_{s=1}^t \bar{g}_s. In fact, this is an important claim because then the dual averaging step reduces to an operation with gradient estimates \bar{g}_1,...\bar{g}_T.

I simply cannot see how this is true. I feel this is basic algebra. First of all, even for the equality to hold, we must have \eta \sum_{s=1}^t \bar{g}_s (I can understand missing \eta is a typo). However, even with that, I dont think the equality holds. From definition, it can be seen that \eta \sum_{s=1}^t \bar{g}_s=

\sum_{s=1}^t \eta \frac{t-s+1}{K+1} \hat{g}_s. Thus, for each \hat{g}_s, the coefficient associated with it is :\eta \frac{t-s+1}{K+1}. However, for all s, the coefficient is not equal to \alpha_{s,t}. By definition, \alpha_{s,t}=\eta \frac{t-s+1}{K+1} only for s \> t-k. So, how is the equality holding when s\le t-k?

Lemma 11: (8c)\le 0. When i looked at the proof of this statement, I once again felt there was a mistake. It is stated in the proof that sum of the \bar{f}_t(x^*) terms is equal to sum of the \hat{f}_t(x^*) terms. I dont see how that is true. This can be simply tested by taking T=2 and K=1 (allowed by condition of Thm. 9) By definitions, for T=2 and K=1, \sum_{t=1}^T \bar{f}_t(x^*)= \hat{f}_1(x^*) + (1/2) \hat{f}_2(x^*), which is obviously not equal to \hat{f}_1(x^*) + \hat{f}_2(x^*).

I did not even look through the rest of the proofs. I find the two mistakes very surprising since they seem to be basic algebra. It is of course possible that the authors made some typos which have completely changed some important definitions. Otherwise, I will be happy to be corrected.

In terms of presentations, I have a few small questions: 1. In abstract, in line 026, it should be that the regret guarantee is for smooth convex functions, not all convex functions. For the general Lipschitz BCO problem, the lower bound might still be \Omega (T^{2/3}) (or even higher, since the best known result is Flaxman's O(T^{3/4}). 2. I am not sure what Thm.5 means (or rather, the definition of K_{y,\epsilon}. In the definition of K_{y,\epsilon}, there is an \alpha term present and no epsilon term. Is \alpha= \epsilon? It seems to be a typo. 3. Thm 9. In the regret bound, the term \epsilon appears. Nowhere in the algorithm or in the statement of Thm.9 do we have \epsilon. I understand that \epsilon probably appears while using an upper bound on the self-concordant R(), but this is quite poor in terms of presentation. Please modify the statement of Thm 9 to explicitly mention \epsilon. Or dont mention \epsilon at all and use T (like in Saha and Tewari).

4. In Saha and Tewari's paper, the authors dont analyze regret directly in terms of x^*. They instead consider a term \bar{x}, which is 1/\sqrt(T) away from boundary of K. This allows R(\bar{x}) \le 2 \new log(T). How do you use x^* directly?

5. Please provide definition of Minkowski functional in supplement to make the paper self-containing. Specifically, I would like to know what is the Minkowski's func. associated with a Dikin's ellipsoid. Like specifically what does the functional look like for Dikin's ellipsoid?

Update: It appears that the proofs are correct. I accept the paper.
Summary: The paper is straight forward in terms of accept/reject decision. It is a purely technical paper, promising a better regret bound for the well studied BCO problem, with smooth functions. So, if the proofs are correct, the paper should be interesting for people interested in the BCO problem. I have found, what I believe, are important mistakes in the paper and that's why I am rejecting it. I will definitely accept the paper if it is actually my mistake rather than mistakes in paper.

Submitted by Assigned_Reviewer_3

The paper improves on the state of the art upper bound on the regret of adversarial smooth function minimization in the bandit setting. In particular, the authors show that by averaging past gradients in some fixed-size window, one can lower to variance of the gradient estimates generated by the algorithm of Saha & Tewari. Thus, the regret bound is improved from O(T^{2/3}) to O(T^{5/8}).

This result is interesting since it refutes the conjecture that the minimax regret rate in this setting is O(T^{2/3}), and strengthens the possibility that the lower bound of Omega(T^{1/2}) is tight.

The writing is very good.

The results are novel and well-presented. Although the improvement over past results is small, its impact is significant. Thus, in my opinion, this paper should be accepted.

Additional comments =================== - Line 148: v should be sampled from the unit ball and not from the sphere. v is sampled from the unit sphere for the calculation of the gradient but not the perturbed function value (Stoke's theorem). - Line 188: \|z\|_x must be strictly less than 1. - Line 322: Missing eta in the RHS of the equation.
Summary: The paper improves on the state of the art upper bound on the regret of adversarial smooth function minimization in the bandit setting. An excellent paper and should definitely be accepted.

Author Feedback
Author rebuttal: We thank the reviewers for their helpful comments. We address specific points raised in the reviews:

Reviewer 2:
===========

* Possible mistakes in Eq. on Line 322-323, and in proof of Lemma 11.

We rechecked our derivation and verified its correctness. As you pointed out, we are missing \eta in line 322-323, but otherwise everything seems to be correct - see details below. In the final version, we will elaborate on why the equation is correct---the computation might be confusing indeed.

Before we address the concrete concern, we point out that even if the equation were wrong, our main result would still be intact. This equation simply establishes that dual averaging with the alpha-weighted \hat{g}_t vectors is equivalent to dual averaging with uniformly weighted \bar{g}_t vectors. Namely, it merely gives us a way to unify the pseudocodes of our algorithm and the previous algorithms. The algorithm that we analyze is (unweighted) dual averaging with \bar{g}_t vectors, and the definition of \alpha does not play a critical role in our analysis.

* "From definition, it can be seen that \eta \sum_{s=1}^t \bar{g}_s = \sum_{s=1}^t \eta \frac{t-s+1}{K+1} \hat{g}_s."

This is incorrect: notice that in the sum \sum_{s=1}^t \bar{g}_s, each \frac{1}{k+1} \hat{g}_s term is summed up exactly k+1 times, except for the last k terms (corresponding to s=t-k+1,...,t-1,t). Thus, the coefficient of each \hat{g}_s term is precisely \eta, except for the last k terms whose coefficients are \eta \frac{t-s+1}{k+1}. This is exactly the definition of the alpha_{s,t} coefficients in Eq.5.

* "It is stated in the proof that sum of the \bar{f}_t(x^*) terms is equal to sum of the \hat{f}_t(x^*) terms."

In fact, in the proof of Lemma 11 (see lines 731-734) it is stated that the first sum can only be smaller than the second, and is not necessarily equal to it. This is indeed the case in your test example with T=2 and k=1.

* Other comments: thank you for spotting these typos/inconsistencies; they will all be fixed in the final version.

Reviewer 5:
===========

* "The paper marginally improves a known regret bound"; "techniques do not seem convincing for fully closing the gap between known lower and upper bounds".

The BCO problem is a difficult and fundamental open problem in online learning. Progress on such problems is often achieved one step at a time. The previous papers on this topic were also incremental steps, but we are glad that they were published because we were able to build our result on top of them. Similarly, we hope that others will build on our result and take additional steps towards the resolution of the open problem. We think we did more than just marginally improve the bound: we proposed a new algorithmic ingredient and presented its nontrivial analysis. We agree that closing the gap will probably require heavier machinery, but at the same time feel that our results are a non-trivial step that the NIPS audience will be interested to hear about.